# Behavioral Response of Bean Goose (*Anser fabalis*) to Simulated Ship Noises at Lake

**DOI:** 10.3390/ani12040465

**Published:** 2022-02-14

**Authors:** Sujuan Wang, Lizhi Zhou, Jinzhou Cai, Bo Jiang, Wenbin Xu

**Affiliations:** 1School of Resources and Environmental Engineering, Anhui University, Hefei 230601, China; x18301089@stu.ahu.edu.cn; 2Anhui Province Key Laboratory of Wetland Ecosystem Protection and Restoration, Anhui University, Hefei 230601, China; 3Changjiang Water Resources Protection Institute, Wuhan 430051, China; jzcai2010@hotmail.com (J.C.); jbshuibao415@126.com (B.J.); 4Key Laboratory of Ecological Regulation of Non-Point Source Pollution in Lake and Reservoir Water Sources, Changjiang Water Resources Commission, Wuhan 430051, China; 5Management Department of Anhui Shengjin Lake National Nature Reserve, Chizhou 247210, China; xuwbvip@163.com

**Keywords:** ship noise, disturbance distance, behavior response, bean goose, waterbrids

## Abstract

**Simple Summary:**

Environmental noise influences the behavioral patterns of animals. However, few quantitative studies have evaluated the effects of ship noise on wintering waterbirds in lakes. In this study, the effects of ship noise simulated by noise playback at different intensities and interference distances on the behaviors of the bean goose, a wintering waterbird species, were evaluated. Sensitivity to noise was higher in small populations than in large populations. Noises of >70 dB at distances of <100 m and >80 dB at <200 m clearly altered the flight patterns of bean geese. This study provides insight into the tolerance of endangered and protected waterbirds to environmental noise and may guide the development of strategies to minimize the impact of ship noise.

**Abstract:**

Wild animals are vulnerable to environmental noise. In wetlands, wintering waterbirds are easily disturbed by ship noises; however, the behavioral changes of waterbirds in response to different levels of noise are unclear. We simulated the acoustic environment created by ship movement to investigate the effects of ship noise on foraging, vigilance, and flight behaviors of the wintering bean goose (*Anser fabalis*). In particular, we used a noise playback method to simulate the acoustic environment created by ship operations at various noise levels (i.e., background noise <50 dB, 60, 70, 80, 90, and 100 dB), distances from the noise (i.e., short <100 m, medium 100–200 m, and long distances 200–300 m), and noise duration (i.e., short 0–1 min, medium 2–3 min, and long 4–5 min). Results indicated that the noise intensity and interference distance had obvious influence on the bean geese behavior, but the noise duration had no effect. Smaller populations (*N* ≤ 30) were more sensitive to noise interference. As the noise level increased, the frequency of foraging behavior decreased and the frequencies of vigilance and flight behaviors increased, particularly above 70 dB. For noises >70 dB at short disturbance distances and >80 dB at medium disturbance distances, flight behavior increased significantly. These findings suggested that ships should keep a distance of more than 200 m from waterbirds to reduce noise interference.

## 1. Introduction

Noise is a very common form of disturbance with harmful effects on many species, e.g., causes stress responses of animal behaviors [1,2], reducing foraging efficiency, and social communication [3,4]. To minimize noise interference, animals can change temporal or spatial patterns of behavior [5,6], change the intensity of behaviors, or adopt alternative behaviors [7].

Behavior is a common indicator of the tolerance of birds to noise disturbance [8,9]. Noise can affect the availability of sound information to birds and alter foraging and vigilance [10]. For example, in captivity, zebra finches (*Taeniopygia guttata*) spend more time on vigilance in noisy areas than in quiet areas when feeding, resulting in a low foraging efficiency [6]. In response to simulated traffic noise, when at 80 dB, the thrushes (*Garrulax canorus*) showed obvious retreat behavior, then gradually distributed to the region with low noise intensity [11]. Most of these studies have been conducted in laboratories with caged birds, thus this approach may ignore important ecological factors and thereby the results may not be generalizable to field conditions. Meanwhile, through field experiments, when simulating railway noise the black-necked crane (*Grus nigricollis*) gradually moves away from the noise source at approximately 60 dB of noise, with an escape distance of approximately 60–80 m from the noise source [12]. In addition, highway noise causes avoidance behavior of black-necked cranes, and the average avoidance distance is about 135.18 m; the closer to the highway, the easier it is to be startled [13]. Further, when cars drive by at a nearer distance of 150 m from the road, mallard (*Anas platyrhynchos*) are severely affected by the noise and show flight behavior, while at a farther distance of 360 m, swan geese (*Anser cygnoides*), bean geese (*Anser fabalis*), and grey crane (*Grus grus*) were only slightly disturbed [14]. There are also research findings that show the influence area of road noise on different birds was as far as 189.63 m, but the avoidance distance of birds had no relationship with population size. However, there are other findings that demonstrate that population size has an effect on the vigilance behavior of birds. For example, when there is no noise, the proportion of vigilance individuals of hooded crane (*Grus monacha*) decreased with population increase. Moreover, when hooded crane population sizes increased 90 individuals, the percentage of vigilance individuals dropped to a minimum, and remained essentially unchanged [15]. Most of these studies focused on field observations and research conducted on roads and railways on the behavior of birds, or combined with habitat change, population size, human interference, and other forms of interference [16]. However, few studies have evaluated bird behavioral responses to ship noise. In particular, quantitative studies of the influence of noise intensity on the behaviors of wintering waterbirds based on simulated ship noise are lacking. At present, many wintering waterbirds depend on the wetlands, since the disturbance caused by ship noise is affecting their habitat and living environments. Therefore, it is necessary to investigate the ecological impact of ship noise on overwintering waterbirds.

Recent research has shown that container ships, oil tankers, ferries, and bulk carriers are the biggest sources of shipping noise at present and in future vessel traffic services [17,18]. Some ship equipment can emit a high level of airborne noise [19]. On a typical ship, there are ubiquitous 24 h-day high noise environments, with noise levels ranging from 87 to 102 dB [20]. Noise pollution from ships with frequent traffic, such as fishing and patrol ships [21,22], has become one of the main types of environmental pollution in rivers, lakes, and other water areas, and has gradually become one of the main hazard sources that harm the water environment and destroy the water ecosystem. Therefore, it is necessary to conduct simulated ship noise experiments to evaluate the impact of ship noise pollution by observing the behavior patterns of overwintering waterbirds.

Bean geese are a common waterbird species wintering in the lakes of the middle and lower Yangtze River floodplain with the largest populations and widest distribution [23]. Their habitats are mainly in the shallow waters of lakes, lakeside meadows, and farmlands and are often mixed with other swimming birds (e.g., spot-billed ducks (*Anas zonorhyncha*) and mallards), waders (e.g., little egrets (*Egretta garzetta*), and herons (*Ardea cinerea*)). In this study, we focused on its behavioral responses to ship noise with the hope of providing a scientific basis for shipping management. We simulated ship noise and observed the effects of the ship noise source, distance, and duration on the behaviors of bean geese. We then analyzed the relationships between noise-related parameters and the flight and other behaviors. Finally, the lowest noise intensity and distance were determined, and the ecological impact of noise on waterbirds was evaluated.

## 2. Materials and Methods

### 2.1. Ethics Statement

The ship noise simulation experiment did not involve bird catching or hunting. Approval was obtained from the local wildlife protection departments. The research process complies with current Chinese laws.

### 2.2. Study Site Selection

The Shengjin Lake National Nature Reserve (30°15′–30°30′ N, 116°55′–117°15′ E) in Anhui Province is an important gathering area for waterfowl in the middle and lower Yangtze River floodplains [24,25]. Every October, a large number of bean geese arrive at the Shengjin Lake for the winter seasons. The number reaches the annual maximum in mid-early December and decreases from March to April of the following year. With the approval of the National Reserve Management Office, the research site was the farmland area of the Lianhe Village in the experimental area of the reserve. The study area had a large number of shrubs up to 1 m, providing good cover conditions for noise playback equipment and observers. The site was far from residential areas, with no human activities, vehicle traffic, and other disturbances. Daytime ambient noise met level one acoustic environmental quality standard (55 dB).

### 2.3. Typical Ship Noise Recording and Playback Design

To obtain ship noise for the experiment, a Philips VIR8800 recording pen (Shenzhen Jinghua Electronics Co., LTD, Shenzhen, China) and a high-precision noise tester (AS844+) (Shenzhen Xima Yinghao Trading Co., LTD, Shenzhen, China) were used to monitor the noise of 1000 t ships on Nanfei River, Hefei city, Anhui Province. The equivalent sound level of 5 min at 1 m distance from the exhaust cylinder was recorded synchronously by noise meter. The test method refers to the HJ 640-2015 “Technical Specification for Environmental Noise Monitoring Routine, Monitoring of Urban Environment Noise”. According to the field measurement, the noise level at 1 m of the upstream full load ship of 1000 t class was 97.4 dB; therefore, the noise was selected as the noise source for simulations. Adobe Audition CS6 audio editing software (Adobe Systems Incorporated, San Jose, CA, USA) was used to select the typical time of noises, in addition to exhaust cylinder noise, and to eliminate other sudden sounds, such as whistles. When listening through headphones, the recording was rechecked and 2 h of noise was edited to simulate the noise in the field experiment.

### 2.4. Distance between the Noise Source and the Distribution of Bean Geese

Bean geese remain mostly in the paddy field or forage on the beach. When observers approached, they flew away. Therefore, the principle of triangulation was adopted to calculate the nearest distance between the animals and noise source (Figure 1). The three vertices were bean geese distribution area (point A), noise equipment (point B) and reference point (point C). An electronic Total Station was used to measure the two interior angles, *α* and *β*, of vertex B and C in the triangle, and measure the distance L between point B and point C, as well as calculate the distance, D, between point B and point A. By using this method, noise equipment could be arranged in the area far from the distribution area of bean geese, which could avoid the influence on the waterbirds as much as possible and meet the requirements of this experiment. The calculation formula is as follows: $$D=L\times \frac{\sin\beta }{\sin(\alpha +\beta )}$$

### 2.5. Noise Disturbance and Behavioral Observations

A preliminary experiment was conducted from December 9 to 11, 2019. Frequent activity areas of bean geese were first identified and the main behaviors during the wintering period were observed. We found that if there was no human disturbance, the foraging and resting behavior of bean geese generally began in the morning and continued into the evening in the same area. If disturbed by traffic, honking, or pedestrians, the geese would stay away from, or fly out of, their original area. By simulating ship noise in the field, when the noise level reached 100 dB and the horizontal distance was more than 300 m, the researchers heard the noise below the background value (dB < 50).

The formal experiment was conducted from December 12 to 28, 2019. Under normal conditions, the observation time was 8:30 to 17:30, or when the bean geese flew away from the grassy beach. Before the experiment, an area with a high concentration of bean geese was selected, and the population size was in the range of 10–60, far from areas with substantial human interference, such as village roads, to ensure that the subject was only disturbed by the noise of a single ship noise, and no other disturbances. The noise equipment SAST/A60 (Shenzhen Xianke Enterprise Group, Shenzhen, China) was placed in a suitable position near the typical area, and the ship sound was oriented towards the study area. Environmental background noise was measured in the test area. Owing to the reduced noise sources in the study area in the winter, the equivalent sound level was measured for 5 min, revealing that the background noise in the test area was lower than 50 dB. Noise playback [26,27,28] was used to simulate the sound of ship operations. Six noise level gradients were set: control group (ZR, background noise) and experimental group (60, 70, 80, 90, and 100 dB) obtained by adjusting the volume of the audio equipment 1 m away from the playback device. The behavior observation time was 30 min (i.e., continuously playing 6 gradients of noise for 5 min each) or the time taken for all observed objects to exhibit flight, and the observations were made at three equal intervals in each noise period (i.e., at 0, 2, and 4 min). In the experiment, the observers and the noise equipment were hidden, and kept a certain distance from the subjects with no contact. Before playing the noise, relatively concentrated bean geese were selected to ensure that all geese could be observed. Then, the instantaneous scanning method was used to ensure that the behavioral responses of all experimental subjects could be observed. Binoculars (SWAROVSKI EL 8.5 × 42) (Swarovski (Shanghai) Trading Co., LTD, Shanghai, China) and monoculars (SWAROVSKI ATS80HD+20-60) (Swarovski (Shanghai) Trading Co., LTD, Shanghai, China) were used to observe and record the six types of behaviors in bean geese [29]: foraging (head lowering, moving slowly, pecking at ground food, approaching or stopping at the water surface, and drinking several times), grooming (bending neck toward the back, moving beak back and forth to sort out feathers), rest (bending legs together, sticking chest and abdomen on the ground or the water surface, neck shrinkage), moving (a body displacement, two legs moving frequency to speed up, moving over a long distance in a short time), warning (neck elongation, head scanning the surrounding environment or watching the interference source), flight (panic, constantly emitting cries, flapping wings quickly, spreading wings, flying in the opposite direction to the interference source or moving over a long distance and stopping). The observation time was 30 min when six noise level gradients were played, or all observed objects had time to exhibit flight. After the experiment was completed, a new population of bean geese was observed following the same methods.

The disturbance distance was divided into short (<100 m), medium (100–200 m), and long distances (200–300 m). In actual observations, noise transmitted to the bean geese at greater than 300 m was close to the level of natural noise. The duration of interference was divided into three stages: T1 (0–1 min), T2 (2–3 min), and T3 (4–5 min). The population sizes of bean geese were set to *N* (0 < *N* ≤ 60), *N1* (*N* ≤ 30), and *N2* (30 < *N* ≤ 60) [29]. In the process of recording bean geese behaviors, data for other sources of interference were eliminated, such as vehicles, horns, and barking. In the natural environment, the activity range of bean geese is large, and they may disperse. Therefore, groups with good visibility and high concentrations were selected as the observation objects. Individuals exhibiting each behavior were counted as follows:$$X=\frac{n}{N}\times 100\%$$where X represents the proportion of bean geese exhibiting a certain behavior, *n* represents the number of bean geese exhibiting a certain behavior, and *N* represents the total number of bean geese observed.

A total of 30 noise simulation experiments were carried out, and a total of 495 behavioral observations were collected (population size *N*), including 251 (population size *N1*) and 244 (population size *N2*).

### 2.6. Statistical Analysis

SPSS 24.0 was used for all data processing and statistical analyses. The K-S test was used to verify whether the data conformed to a normal distribution. The number of bean geese population was a fixed factor. Then, univariate ANOVA and multivariate ANOVA in the general linear model were used to analyze the percentage of individuals exhibiting each behavior (foraging, vigilance, and flight) and the relationships with the noise intensity and disturbance distance, as well as their interaction with different population size. The SNK (Student-Newman-Keuls) test was used to compare the differences between the pairs of influencing factors and the behaviors of bean geese.

## 3. Results

### 3.1. Relationship between Foraging Behavior and Influencing Factors

General linear model analysis of univariate ANOVA was used to evaluate foraging behavior with respect to noise intensity, interference distance, and interference duration. The model test results were available, and it was found that when the population size was *N*, there was a significant difference of noise intensity on foraging behavior (F = 6.302, *p* < 0.001, df = 5). When for *N1*, there also was a significant difference (F = 12.899, *p* < 0.001, df = 5). When for *N2*, there was no significant difference (F = 1.441, *p* = 0.222, df = 5). Meanwhile, there was no significant difference of interference distance (F = 0.024, *p* = 0.976, df = 2) and interference duration (F = 2.091, *p* = 0.102, df = 2) on foraging behavior.

It was concluded that noise intensity had influence on the foraging behavior of the small population *N1*. The SNK test is shown in Figure 2. It could be seen that when the population size was *N*, the frequency of foraging behavior was its lowest at 100 dB (i.e., approximately 30.41%), at 70, 80, and 90 dB it was moderate (i.e., 49.93%) for only half of the population, and at ZR and 60 dB was the highest (i.e., 60.42%), which was twice of that for 100 dB. When the population size was *N1*, the frequency of foraging behavior was at its lowest at 100 dB (i.e., 30.53%), while at 80 and 90 dB it was moderate (i.e., 42.88%) for only half of the population, and at ZR, 60, and 70 dB it was highest (i.e., 60.42%), which was twice of that for 100 dB. However, *N2* had no significant effect on foraging behavior. Therefore, with an increase in noise, the foraging ratio lower at *N1* (smaller populations) was more obvious than that at *N2* (larger populations).

### 3.2. Relationship between Vigilance Behavior and Influencing Factors

General linear model analysis of univariate ANOVA was used to evaluate vigilance behavior with respect to noise intensity, interference distance, and interference duration. The model test results were available, and it was found that when the population size was *N*, there was a significant difference of noise intensity on vigilance behavior (F = 10.431, *p* < 0.001, df = 5). When for *N1*, there was a significant difference (F = 18.516, *p* < 0.001, df = 5). When for *N2*, there was also a significant difference (F = 2.64, *p* = 0.031, df =5). The SNK test is shown in Figure 3. The model test results were available, and it was found that when the population size was *N*, there was a significant difference of interference distance on vigilance behavior (F = 5.378, *p* = 0.006, df = 2). When for *N1*, there was a significant difference (F = 8.48, *p* < 0.001, df = 2). When for *N2*, there was also a significant difference (F = 3.697, *p* = 0.03, df = 5). The SNK test is shown in Figure 4. Meanwhile, there was no significant difference of interference duration on vigilance behavior (F = 0.228, *p* = 0.776, df = 2).

General linear model analysis of multivariate ANOVA was used to evaluate vigilance behavior with respect to noise intensity and interference distance. The model test results were available, and it was found that there was no significant difference of noise intensity and interference distance on vigilance behavior (F = 0.678, *p* = 0.744, df = 10).

It can be seen that when the population size was *N*, the frequency of vigilance behavior was at its highest at 100 dB (i.e., 18.15%), while ZR and 60 dB was at its lowest (i.e., 5.96%), which was one third of that for 100 dB. When the population size was *N1*, the frequency of vigilance behavior was highest at 100 dB (i.e., 18.51%), while at ZR it was lowest (i.e., 5.92%), which was one third of that for 100 dB. When the population size was *N2*, the frequency of vigilance behavior was highest at 100 dB (i.e., 18.66%), while at ZR it was lowest (i.e., 7.04%), which was half of that for 100 dB. It could be seen that when the population size was *N*, the frequency of vigilance behavior was highest at close and medium distances (i.e., 12.67%), and at long distances was at its lowest (i.e., 5.96%), which was half of that for close distances. When the population size was *N1*, the frequency of vigilance behavior was highest at close and medium distances (i.e., 12.99%), and at long distances was at its lowest (i.e., 8.44%), which was half of that for close distances. When the population size was *N2*, the frequency of vigilance behavior was at its highest at close and medium distances (i.e., 14.11%), and at long distance was at its lowest (i.e., 6.23%), which was half of that for medium distances.

### 3.3. Relationship between Flight Behavior and Influencing Factors

General linear model analysis of univariate ANOVA was used to evaluate flight behavior with respect to noise intensity, interference distance, and interference duration. The model test results were available, and it was found that when the population size was *N* there was a significant difference of noise intensity on flight behavior (F = 24.389, *p* < 0.001, df = 5). When the population size was *N1*, it had a significant difference (F = 43.58, *p* < 0.001, df = 5). When the population size was *N2*, it also had a significant difference (F = 8.178, *p* < 0.001, df = 5). The SNK test is shown in Figure 5. The model test results were available, and it was found that when the population size was *N*, there was a significant difference of interference distances on flight behavior (F = 3.909, *p* = 0.022, df = 2). When the population was *N1*, it had significant difference (F = 9.78, *p* < 0.001, df = 2). When the population was *N2*, it had no significant difference (F = 0.362, *p* = 0.697, df = 5). The SNK test is shown in Figure 6. Meanwhile, there was no significant difference of interference duration on flight behavior (F = 3.427, *p* = 0.089, df = 2).

It can be seen that when the population size was *N*, the frequency of flight behavior was highest at 100 dB (i.e., 33.76%), at 80 and 90dB were medium (i.e., 20.42%), and at ZR and 60 dB were lowest (i.e., 0.00%). When the population size was *N1*, the frequency of flight behavior was highest at 100 dB (i.e., 33.77%), at 80 and 90dB were medium (i.e., 20.42%), and at ZR was lowest (i.e., 0.00%). When the population size was *N2*, the frequency of flight behavior was highest at 100 dB (i.e., 39.54%), at 70, 80, and 90dB were medium (i.e., 24.32%), and at ZR was lowest (i.e., 0.00%). It also can be seen that when the population size was *N*, the frequency of flight behavior was highest at close distances (i.e., 15.00%), while at long distance was lowest (i.e., 8.28%), which was half of that for close distances. When the population was *N1*, the frequency of vigilance behavior was highest at close distances (i.e., 16.31%), while at long distance was lowest (i.e., 6.72%), which was half of that for close distances. However, when the population was at *N2*, there was no significant effect on flight behavior. Therefore, with an increase in noise, the flight ratio was higher at *N1* (a smaller population) and was more obvious than that at *N2* (larger populations).

General linear model analysis of multivariate ANOVA was used to evaluate flight behavior with respect to noise intensity and interference distance. The model test results were available, When the population size was *N*, there was a significant difference of noise intensity and interference distance on flight behavior (F = 2.555, *p* = 0.007, df = 10). When the population was *N1*, there was a significant difference of noise intensity and interference distance on flight behavior (F = 5.923, *p* < 0.001, df = 10). When the population was *N2*, there was no significant difference of noise intensity and interference distance on flight behavior (F = 1.887, *p* = 0.063, df = 10). It can be seen that for noises >70 dB at short disturbance distances and >80 dB at medium disturbance distances, the flight behavior increased significantly, and with the increase of noise flight frequency gradually increased (Figure 7). However, there was no significant change in flight behavior at long distance.

## 4. Discussion

Noise can affect the availability of sound information to birds and alter their foraging and vigilance behaviors [10]. Our research found that, in the natural environment, one or two bean geese in each population were responsible for guarding, while the others foraged or rested on the grassland. This is consistent with other studies, where, under no noise, the most frequent behavior was foraging (63.8%), followed by resting (15.6%), and other behaviors (such as alarm and flighting, standing, and walking) [30]. When the simulated noise was 60 dB, the distribution of behaviors was similar to that of bean geese without noise. Up to 70 dB, we observed that vigilance is always observed in bean goose populations, and as the number of individuals increases, foraging efficiency decreases [31], suggesting that noise may be sufficient to cause a stress response in birds [4]. Further, at this noise level, very few individuals showed retreat or flight behaviors, indicating that bean geese have the ability to adapt to or tolerate mild noise stimulation. When the intensity is 75 dB, birds showed an increased vigilance and reduced foraging behavior [26,32]. Moreover, as the intensity increased to 80 dB, the number of individuals showing foraging behavior decreased gradually, while the frequency of vigilance behaviors continued to increase. Some individuals moved away from the observed population, resulting in retreat or flight. A former study has shown that the thrush shows greater retreat behavior at 75.4 dB than 80 dB [11], and this may be due to the fact that the birds were raised in captivity, with a different living environment from the natural environment. However, at our field site, the noise was attenuated, to a certain extent, at various distances; accordingly, the noise intensity resulting in flight was higher than the actual noise intensity.

Noise can keep birds away from certain sites. The initial proximity to humans can affect birds, resulting in flight [10]. Vehicle noise within 100 m of a road has a great impact on bird behavior, of which flight (retreat) accounts for 11% of behaviors [33]. We found that ship noises at close distances (<100 m) have a greater effect on the behavior of bean geese than noises at longer distances, resulting in reduced foraging activities and increased time spent on guard. The noise source was continuous and fixed, which may result in a longer-term effect on birds than those of moving sources, such as vehicles. Studies have shown that black-necked cranes maintain distances of 135.18 m from highways [12] and mallards are seriously affected within 150 m from roads [14]. At medium distances (100–200 m), bean geese continued to feed on grass; however, vigilance behavior increased. A portion of individuals exhibited obvious preparation for flight, while others maintained foraging behavior or remained vigilant, revealing that there is intraspecific variation in noise tolerance [34]. Unlike individuals with high tolerance to noise, those with low tolerance will identify weak stimuli as risks and show behavioral responses [35]. In addition, there are differences between waterbird species in their tolerance to noise [32]. In experiments, noise not only directly affects target subjects, but also unintentionally affects other organisms in the ecosystem [36,37]. For example, playing noise can cause agitation among livestock in nearby residential areas, and the results of the experiment can be affected by the indirect effects of interactions with other species, particularly in mixed populations of great white-fronted geese (*Anser albifrons*), tundra swans (*Cygnus columbianus*), and hooded cranes. Christoph et al. [21] observed this phenomenon, in which swan geese were frightened by barking and flew to other habitats, resulting in a significant difference in the distribution between conditions with barking and no barking (*p* < 0.001). However, with an increase in the distance from the noise source, the number of bean goose individuals exhibiting foraging behavior within 200 m remained basically unchanged, while the individuals exhibiting vigilance behavior showed a slight decrease and no or very few individuals exhibited flight. Bean geese and greylag geese (*Anser anser*) [14] located 360 m from the highway are also only slightly disturbed by noise.

The flight distance of birds increases as noise interference increases [38]. In the Haizhu Lake area, Vervet spoonbills (*Platalea minor*) produced flight at a noise value of 61.6 ± 5.2 dB when disturbed by humans [39]. When disturbed by simulated noise in farmland habitats, the bean geese exhibit flight reached 70 dB. It is possible that Vervet spoonbills in the Haizhu Lake area are more prone to flight in response to various disturbances, such as noise and human activities. Research also suggests that hooded cranes show vigilance behaviors in response to noise and move away from the noise sources [15], as well as when ambient noise increases by approximately 60 dB, black-necked cranes gradually escape to the side to a distance approximately 60–80 m away from the noise source [13]. They are more sensitive to human interference and less vigilant to traffic noise, such as motor vehicles or ships [13]. Bean geese live in a natural environment, avoiding interference from human activities, and are sensitive to noise. In a study of the impact of highway noise, for noises >60 dB at short distances (<100 m) and noises <60 dB at long distances (>300 m) on bird diversity indicated that the richness and encounter rate of birds in close range decreased significantly and the species composition was significantly different [40]. Our results also indicated that 70 dB at a short distance and 80 dB at a medium distance result in the departure of some individuals from the observed group of bean geese, and they began to move away from the noise source until they retreated to a safe distance to forage and rest, while at a long distance, the population and behavior of bean geese did not change significantly. The noise value measured in this study was relatively high when the geese exhibited flight. The difference among studies may be related to the indirect influence of vehicle traffic and human disturbances [21], the group size of social foragers [6], or a difference in sensitivity to noise among species [11].

## 5. Conclusions

With an increase in noise, the frequency of foraging behavior of bean geese wintering at Shengjin Lake decreased, while the ratio of vigilance to flight behaviors increased. Additionally, sensitivity to noise was higher in small populations (*N* ≤ 30) than in large populations. For 70 dB at a short distance and 80 dB at a medium distance, the frequency of flight increased significantly; accordingly, these values can be considered thresholds for the influence of simulated ship noise on the behavior of bean geese. These results suggested that ships should maintain a distance of more than 200 m from waterbirds to reduce noise disturbance. Since different waterbirds have different sensitivities to noise, behavioral research and analyses of more waterbirds should be conducted, in order to establish a corresponding database and provide a scientific basis for the protection of waterbirds. Our study has certain limitations. First, our experiment only considers the noise intensity, not the frequency. Second, the noise duration in our simulations was not long enough to detect more behavioral changes. These limitations should be clarified in future studies.

## Figures and Tables

**Figure 1 animals-12-00465-f001:**
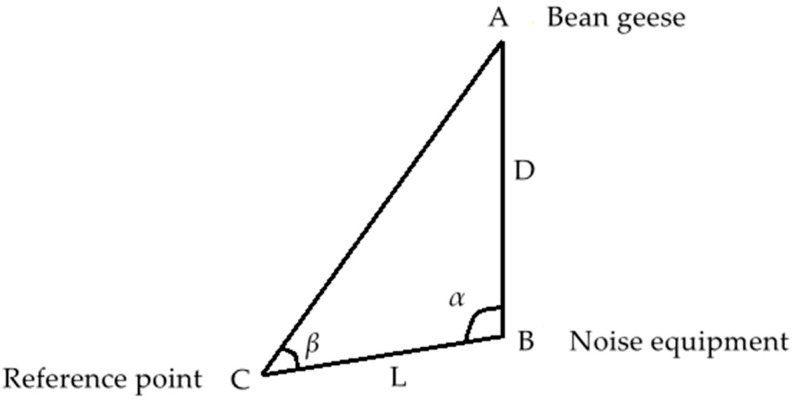
Schematic diagram of the distance between the bean geese and the noise source.

**Figure 2 animals-12-00465-f002:**
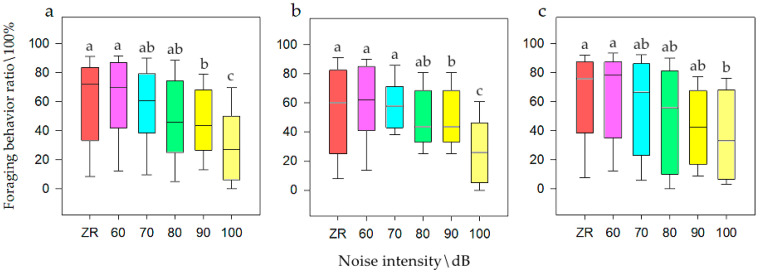
Foraging behavior of bean geese ((**a**) Population *N*; (**b**) Population *N1*; (**c**) Population *N2*) of different noise intensity (ZR (background noise), 60, 70, 80, 90, and 100 dB). Different letters in the graphs represent significant differences from one-way ANOVA and SNK test. Bars represent means; error bars denote standard deviations. The bar with the highest foraging behavior ratio is marked as “a”. The foraging behavior ratio is compared with other noise gradients, and the difference is significant, marked as “b” and “c”.

**Figure 3 animals-12-00465-f003:**
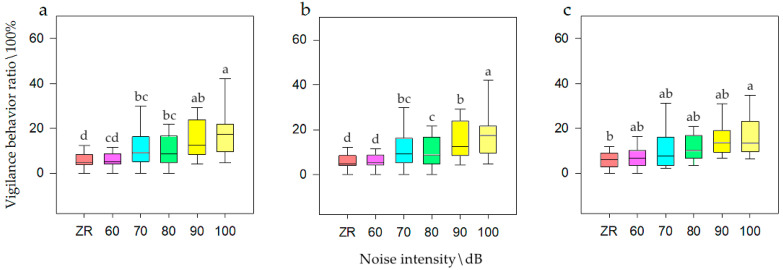
Vigilance behavior of bean geese ((**a**) Population *N*; (**b**) Population *N1*; (**c**) Population *N2*) of different noise intensity (ZR (background noise), 60, 70, 80, 90, and 100 dB). Different letters in the graphs represent significant differences from one-way ANOVA and SNK tests. Bars represent means; error bars denote standard deviations. The bar with the highest vigilance behavior ratio is marked as “a”. The vigilance behavior ratio is compared with other noise gradients, and the difference is significant, marked as “b” and “c”.

**Figure 4 animals-12-00465-f004:**
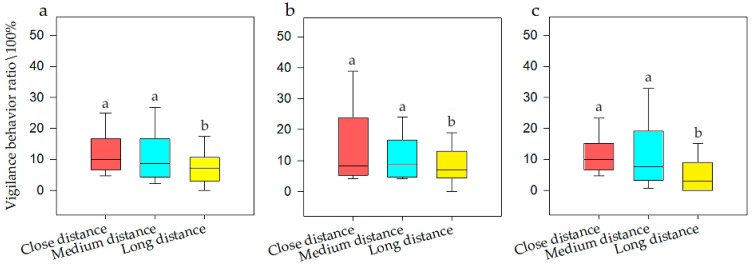
Vigilance behavior of bean geese ((**a**) Population *N*; (**b**) Population *N1*; (**c**) Population *N2*) of close distance, medium distance, and long distance. Different letters in the graphs represent significant differences from one-way ANOVA and SNK tests. Bars represent means; error bars denote standard deviations. The bar with the highest vigilance behavior ratio is marked as “a”. The vigilance behavior ratio is compared with other interference distances, and the difference is significant, marked as “b”.

**Figure 5 animals-12-00465-f005:**
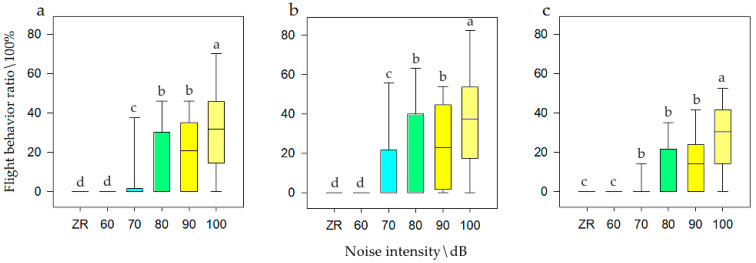
Flight behavior of bean geese ((**a**) Population *N*; (**b**) Population *N1*; (**c**) Population *N2*) of different noise intensity (ZR (background noise), 60, 70, 80, 90, and 100 dB). Different letters in the graphs represent significant differences from one-way ANOVA and SNK tests. Bars represent means; error bars denote standard deviations. The bar with the highest flight behavior ratio is marked as “a”. The flight behavior ratio is compared with other noise gradient, and the difference is significant, marked as “b”, “c”, and “d”.

**Figure 6 animals-12-00465-f006:**
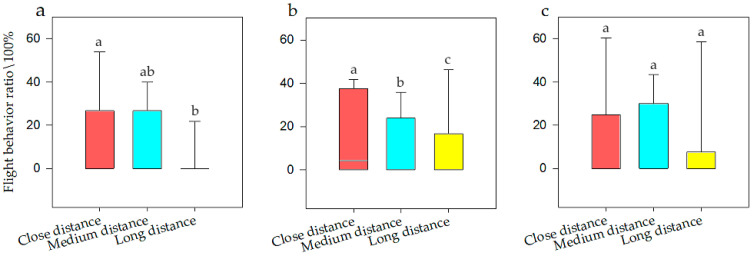
Flight behavior of bean geese ((**a**) Population *N*; (**b**) Population *N1*; (**c**) Population *N2*) of close distance, medium distance, and long distance. Different letters in the graphs represent significant differences from one-way ANOVA and SNK tests. Bars represent means; error bars denote standard deviations. The bar with the highest flight behavior ratio is marked as “a”. The flight behavior ratio is compared with other interference distances, and the difference is significant, marked as “b”.

**Figure 7 animals-12-00465-f007:**
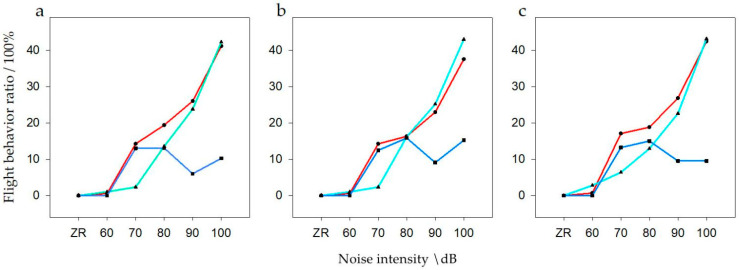
Flight behavior of bean geese ((**a**) Population *N*; (**b**) Population *N1*; (**c**) Population *N2*) of different noise intensity (ZR (background noise), 60, 70, 80, 90, and 100 dB) and interference distances. The red line of black points in the graphs represent close distances; the light blue line of triangular points in the graphs represent medium distances; the blue line of square points in the graphs represent long distances.

## Data Availability

The data presented in this study are available on request from the corresponding author.

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
