# Peer review of "Behavioral Response of Bean Goose (Anser fabalis) to Simulated Ship Noises at Lake"

_animals, 2022, doi:10.3390/ani12040465_

Round 1

Reviewer 1 Report

This paper presents results of a simple experiment of influence of artificial noise on geese behaviour. I like this idea and the results are nice, clear and the whole message is easy to understand with a clear managing implications.

I put some minor comments in the text, most of tchem relate to figures. They need more explanations. Now, a reader would not be able to understand tchem without llooking for details in the methods. It is good to remember that figures and their captions should be self-explaining.

Author Response

Question 1:  L. 37:  Replace "Anser fabalis" with "bean goose".

Reply: According to your comments, "Anser fabalis" has been replaced with "bean goose". See the Keywords, L.37.

Question 2:  L. 72-74: “Ship noise disturbance is one of the main types of pollution in water environments, such as rivers and lakes, and has gradually become one of the main hazards in water ecosystems.”

Reply: Thanks for your advice, I have modified this sentence as follows: “Noise pollution from ships with frequent traffic, such as fishing and patrol [21,22], has become one of the main types of environmental pollution in rivers, lakes and other water areas, and has gradually become one of the main hazard sources that harm the water environment and destroy the water ecosystem. ”. See the Introduction, L.86-89.

Question 3:  L. 78-79: Replace "Bean goose" with “Bean goose”.

Reply: According to your comments, "Bean goose" has been replaced with "Bean geese". See the Introduction, L.92.

Question 4:  ”The site was far from residential areas with no human interference, such as vehicle traffic”.

Reply: According to your comments, I have modified this sentence as follows: “The site is far from residential areas, with no human activities, vehicle traffic and other disturbances”. See the methods, L.117-118.

Question 5:  “In the process of recording bean goose behaviors, data for other sources of interference, such as vehicles, horns, and dogs, were eliminated”.

Reply: According to your comments, I have modified this sentence as follows: “In the process of recording bean geese behaviors, data for other sources of interference were eliminated, such as vehicles, horns, and barking,”. See the methods, L.199-120.

Question 6:  “Figure 2. Effects of noise intensity on foraging behavior of bean geese” .

  Reply: According to your comments, I have modified all the figure in the article. See the methods, L.242-245, L277-280, etc.

Reviewer 2 Report

General comments:

The aim of this article is to examine experimentally the behavioral response of waterbirds to the noise produced by ship traffic (which is a very timely topic). Toward this end, the authors observed the behavior of bean geese in response to the playback of ship engines played at different intensities, distances and for different durations. The authors found that geese foraged less, were more vigilant and initiated flights more frequently when sounds were of higher intensities. Geese were also more vigilant and initiated flights more frequently when sounds were played at close proximity. Interestingly, these results appear to be exacerbated in small groups of birds. Overall, these results could be used as the basis of future guidelines to reduce ship traffic on water bodies where geese spend the winter. Even though I appreciate the experimental approach used by the authors as well as the large amount of behavioral data they were able to collect in the field, I still have several concerns about their study:

-       Many aspects of the experimental design used by the authors are unclear and need to be clarified. For instance, I assume that the recordings of ship noise used for the playbacks did not have a constant intensity. Yet, the authors report a single intensity for each playback they used (does it correspond to the maximal intensity of each recording?). Moreover, no information about the frequency of these recordings is provided while the hearing capacity of birds depend on both the intensity and the frequency of the sounds they are exposed to. The number of ship recordings that was selected for the playbacks is also unclear (only one that was repeated for two hours?). The order of the different acoustic treatments used during playback sessions is also not specified (was it randomized within and among groups?). Finally, it is unclear whether the groups of geese that were monitored were each exposed to a single acoustic treatment or to different acoustic treatments (if so, it should be accounted for in the statistical models).

-       The statistical analyses could be improved and should include “population size” as a fixed factor. For instance, for each dependent variable (foraging, vigilance, flight initiation), I would use a single model including population size, sound intensity, distance and duration as fixed factors. Two-way interactions could also be added. Population ID should also be added as a random factor if the same population was measured multiple times (I am not sure this the case).

-       The overall quality of the writing could be improved.

Specific comments:

Introduction

General comment: Even though the content of the introduction provides relevant information, its structure is very messy. The authors should therefore try to rewrite the introduction more clearly and in  a more structured way. I would suggest the following structure: (1) Noise pollution and its general effects on animals (including birds), (2) The interaction between group size and disturbance, (3) Noise pollution generated by ship traffic, (4) noise pollution on waterbirds.

L. 78: Are there any audiograms from geese (or other waterbirds) that would indicate their hearing sensitivity?

Methods:

L. 103: Replace "should correspond with" with "corresponds to".

L. 134: What kind of disturbance?

L. 148: Is it relevant to use a single intensity to mimic the exposure of geese to ship noise? The noise generated by a ship should indeed be progressively perceived by animals, with increasing intensities when the ship moves closer, highest intensities when the ship is very close, and decreasing intensities when the ship moves away.

L. 151: Were 5-min. recordings played 6 times, resulting in a 30-min. exposure?

L. 155: Were all individual geese within the focal group observed at the same time?

Results:

Table 1: I do not think that these correlations are necessary, as you categorized explanatory variables. Also, replace “Interference time” with “Interference duration”

L. 208-209: What about the effects of noise distance and duration on the foraging behavior of geese?

L. 216-217: Using a single model including population size as a fixed factor (as suggested before) would allow the authors to examine whether this difference is significant.

Figure 2: What are the black dots below and above each bar (minimal and maximal values)?

L. 221: Why was a two-way ANOVA not also used for the foraging behavior of geese?

L. 222: What about the effects of noise duration on the vigilance of geese?

L. 236-237: Again, using a single model including population size as a fixed factor would allow the authors to examine whether this difference is significant.

Discussion:

L. 268: Provide a reference at the end of this sentence.

L. 274: Replace "proved that noise may not be sufficient" with "suggesting that noise may be sufficient" (a higher vigilance is already a sign of stress).

L. 277: Replace "However" with "Moreover"

L. 330: I guess what is meant here with "population decline" is "the departure of some individuals from the observed group of birds" (is it correct?)

Author Response

Introduction:

Question 1: General comment Even though the content of the introduction provides relevant information, its structure is very messy. The authors should therefore try to rewrite the introduction more clearly and in a more structured way. I would suggest the following structure:(1) Noise pollution and its general effects on animals (including birds), (2) The interaction between group size and disturbance, (3) Noise pollution generated by ship traffic, (4) noise pollution on waterbirds.

Reply: Thank you very much for your advice. I have adjusted the structure of the introduction according to your suggestion. First of all,  an overview of the impact of noise pollution on animal; then we analyzed the relationship between the noise intensity and the distance between the noise intensity and the behavior of birds, as well as the response of different populations to the noise disturbance; and then we describe the intensity and impact of noise pollution from ship traffic; Finally, introduces the research purpose, the necessity of studying the impact of ship noise pollution on waterbirds. See Introduction L.52-91.

Question 2: L.78: Are there any audio grams from geese (or other waterbirds) that would indicate their hearing sensitivity?

Reply: At present, in most studies on noise interference and animals. Most of them are described in written language as, when the noise intensity reaches a certain value (e.g. 75, 80, and 100dB, etc.), it will cause influence to the animals (e.g., alert, keep away, flight, etc.).

Methods:

Question 3:  L.103: Replace "should correspond with" with "corresponds to". 

L.134:What kind of disturbance?

Reply: I have already replace "should correspond with" with "corresponds to". See Methods L.105. 

Question 4:  L.134:What kind of disturbance?

Reply: The disturbance refers to human interference (e.g., traffic, honking, or pedestrians, etc.) See the Methods L.139-145.

Question 5:  L.148: Is it relevant to use a single intensity to mimic the exposure of geese to ship noise? The noise generated by a ship should indeed be progressively perceived by animals, with increasing intensities when the ship moves closer, highest intensities when the ship is very close, and decreasing intensities when the ship moves away.

Reply: The experiment simulates the response of a single ship noise to the bean geese. If the geese vigilance or flight due to other disturbances, the group of data will be removed. It has been modified in the paper, see the Methods L.160-164.

Question 6:  L.151: Were 5-min. recordings played 6 times, resulting in a 30-min. exposure?

Reply: The noise of 6 gradients (5 minutes for each gradient) was continuously played for 30 minutes, and the behavioral responses were observed and recorded. In the experiment, the observers and the noise equipment were hidden, and there was no contact with the experimental subjects. It has been modified in the paper, see the Methods L.160-164.

Question 7:  L.155: Were all individual geese within the focal group observed at the same time?

    Reply: Yes, before the experiment, we selected relatively concentrated groups of bean geese in advance to ensure that the behavioral responses of all groups could be observed when the instantaneous scanning method was adopted. It has been modified in the paper, see the Methods L.164-167.

Results:

Question 8:  Table 1: l do not think that these correlations are necessary, as you categorized explanatory variables. Also, replace "Interference time" with "Interference duration”.

Reply: I think your suggestion is correct, and this part has been deleted in the paper.

Question 9:  L.208-209: What about the effects of noise distance and duration on the foraging behavior of geese?

Reply: According to the univariate variance analysis of the general linear model, the disturbance distance and duration did not differ significantly from the foraging behavior of the bean geese. See the Results L.210-219.

Question 10:  L.216-217: Using a single model including population size as a fixed factor (as suggested before) would allow the authors to examine whether this difference is significant.

Reply: The statistical analysis method adopts the general linear model and takes the population number as a fixed factor to analyze the relationship between other influencing factors and the behaviors of bean geese. See the Results L.198-204.

Question 11:  Figure 2: What are the black dots below and above each bar (minimal and maximal values)?

Reply: the black dots below and above each bar, referring to discrete values. But the figure in the article has now been modified.

Question 12:  L.221: Why was a two-way ANOVA not also used for the foraging behavior of geese?

Reply: Two-factor ANOVA of general linear model showed that noise intensity and interference distance had no significant relationship with foraging behavior. See the Results L.210-219.

Question 13:  L.222: What about the effects of noise duration on the vigilance of geese?

Reply: According to the univariate variance analysis of the general linear model, the disturbance duration did not differ significantly from the vigilance behavior. See the Results L.245-246.

Question 14:  L.236-237: Again, using a single model including population size as a fixed factor would allow the authors to examine whether this difference is significant.

Reply: The statistical analysis method adopts the general linear model and takes the population number as a fixed factor to analyze the relationship between other influencing factors and the behaviors of bean geese. See the Results L.198-204.

Discussion:

Question 15:  L. 268: Provide a reference at the end of this sentence.

Reply:  I have appended a reference to this sentence in the article. See the Discussion L.341-343.

Question 16:  L. 274: Replace "proved that noise may not be sufficient" with "suggesting that noise may be sufficient" (a higher vigilance is already a sign of stress).

Reply: According to your comments, "proved that noise may not be sufficient" has been replaced with "suggesting that noise may be sufficient" . See the Discussion L.347.

Question 17:  L. 277: Replace "However" with "Moreover".

Reply: According to your comments, "However" has been replaced with "Moreover". See the Discussion L.351.

Question 18:  L. 330: I guess what is meant here with "population decline" is "the departure of some individuals from the observed group of birds" (is it correct?)

Reply: Yes, your understanding is correct. According to your suggestions, I have modified this sentence. See the Discussion L.405-406.